# Modulation of nucleotide metabolism by picornaviruses

**Lonneke V. Nouwen**[1], **Martijn Breeuwsma**[1], **Esther A. Zaal**[2], **Chris H. A. van de Lest**[2], **Inge Buitendijk**[1], **Marleen Zwaagstra**[1], **Pascal Balić**[3], **Dmitri V. Filippov**[3], **Celia R. Berkers**[2]*, **Frank J. M. van Kuppeveld**[1]*

1 Section of Virology, Division of Infectious Diseases & Immunology, Department of Biomolecular Health Sciences, Faculty of Veterinary Medicine, Utrecht University, Utrecht, The Netherlands, 2 Division Cell Biology, Metabolism & Cancer, Department of Biomolecular Health Sciences, Faculty of Veterinary Medicine, Utrecht University, Utrecht, The Netherlands, 3 Gorlaeus Laboratories, Leiden Institute of Chemistry, Universiteit Leiden, Leiden, The Netherlands

* C.R.Berkers@uu.nl (CRB); F.J.M.vanKuppeveld@uu.nl (FJMK)

**Data Availability Statement:** All relevant data are within the manuscript and its Supporting Information files. The datasets are included in the Supporting Information.

## Abstract

Viruses actively reprogram the metabolism of the host to ensure the availability of sufficient building blocks for virus replication and spreading. However, relatively little is known about how picornaviruses—a large family of small, non-enveloped positive-strand RNA viruses—modulate cellular metabolism for their own benefit. Here, we studied the modulation of host metabolism by coxsackievirus B3 (CVB3), a member of the enterovirus genus, and encephalomyocarditis virus (EMCV), a member of the cardiovirus genus, using steady-state as well as $^{13}$C-glucose tracing metabolomics. We demonstrate that both CVB3 and EMCV increase the levels of pyrimidine and purine metabolites and provide evidence that this increase is mediated through degradation of nucleic acids and nucleotide recycling, rather than upregulation of *de novo* synthesis. Finally, by integrating our metabolomics data with a previously acquired phosphoproteomics dataset of CVB3-infected cells, we identify alterations in phosphorylation status of key enzymes involved in nucleotide metabolism, providing insight into the regulation of nucleotide metabolism during infection.

## Author summary

The family *Picornaviridae* includes many well-known human and animal pathogens. These include the enteroviruses (e.g. poliovirus, coxsackievirus, EV-A71, EV-D68, and rhinoviruses), which cause a variety of diseases ranging from hand-foot-and-mouth disease, myocarditis, and conjunctivitis to aseptic meningitis and acute flaccid paralysis, as well as animal pathogens such as foot-and-mouth disease virus and encephalomyocarditis virus. Upon infection of their host, these viruses modulate several cellular processes for efficient replication and spreading, such as host cell gene expression, intracellular protein and membrane transport, and cell death pathways. However, little is known about the effects of picornaviruses on cellular metabolism. We here show that picornaviruses modulate nucleotide metabolism by inducing nucleic acid degradation and nucleotide recycling, while restricting nucleotide *de novo* synthesis. Insight into picornaviral modulation of

**Funding:** This project has received funding from the Innovative Medicines Initiative 2 Joint Undertaking (JU) under grant agreement n° 101005077. The JU receives support from the European Union's Horizon 2020 research and innovation programme, EFPIA and Bill and Melinda Gates Foundation, Global Health Drug Discovery Institute and University of Dundee. This funding was awarded to FJMvK. Link: https://cordis.europa.eu/project/id/101005077 The funders had no role in study design, data collection and analysis, decision to publish, or preparation of the manuscript.

**Competing interests:** The authors have declared that no competing interests exist.

cellular metabolism is important to increase our understanding of picornavirus-host interactions and may uncover novel therapeutic strategies.

## Introduction

The family *Picornaviridae* comprises a large group of small (~30 nm), non-enveloped viruses with a single stranded positive sense RNA genome (7.5–8 kb) [1,2]. The picornaviridae consist of over 63 genera, which includes enteroviruses, aphthoviruses and cardioviruses. The genus *Enterovirus* includes some well-known human pathogens, including poliovirus, coxsackievirus, echovirus, and several numbered enteroviruses (e.g. EV-A71 and EV-D68), that collectively can cause a variety of diseases ranging from hand-foot-and-mouth disease, myocarditis, and conjunctivitis to aseptic meningitis and acute flaccid paralysis [3]. Moreover, this genus contains the rhinoviruses, which cause the common cold but also exacerbation of asthma and COPD [4]. The genera *Aphthovirus* and *Cardiovirus* contain well-known and important animal viruses with a huge impact on animal health, such as foot-and-mouth disease virus, Theiler's murine encephalomyelitis virus, and encephalomyocarditis virus [5–8].

Picornavirus RNA contains a single open reading frame that encodes a large polyprotein. This polyprotein is autocatalytically cleaved by viral proteases yielding four structural proteins as well as a number of nonstructural proteins involved in viral RNA replication and modulation of host cell functions. Picornaviruses actively modulate host cell processes to ensure efficient replication and to evade innate immune responses. These modulations include the inhibition of cellular gene expression, remodeling of intracellular membranes to create viral replication organelles, disruption of the cytoskeleton, and inhibition of type I interferon and stress signaling pathways [9–13].

In contrast to the extensive modulation of the host cell processes mentioned above, relatively little is known about the remodeling of cellular metabolism by picornaviruses. In general, viruses can actively reprogram the metabolism of glucose, glutamine, fatty acids and nucleotides. This reprogramming may not only ensure the availability of sufficient building blocks for virus replication, but is also implicated in counteracting innate immune responses [14–21]. Thus far, only a few studies have investigated metabolic changes during picornavirus infection [22–25]. Using steady-state metabolomics, these studies point towards differential regulation of glucose, glutamine, lipid and nucleotide metabolism during enterovirus and rhinovirus infection. Here, we describe a comprehensive analysis of both steady-state metabolic changes as well as changes in metabolic pathway activity upon infection of cells with an enterovirus, coxsackievirus B3 (CVB3), and a cardiovirus, encephalomyocarditis virus (EMCV).

We performed tracing experiments with [U-$^{13}$C]-glucose to study the metabolome of infected cells at consecutive time points during infection. Unlike steady-state metabolomics, tracing experiments allow for the deduction of metabolic pathway activity, providing a detailed understanding of the viral effects on metabolic regulation. In addition, we integrated these metabolomic data with the previously acquired phosphoproteomic data of CVB3-infected cells to get more insight in the underlying mechanisms of metabolic alterations during infection. We provide evidence that both CVB3 and EMCV infection increase metabolite levels in the purine and pyrimidine pathways by increasing nucleic acid and nucleotide degradation and the recycling of nucleotides, rather than through increasing nucleotide *de novo* synthesis. Our analyses advances our understanding of how picornaviruses shape cellular metabolism during infection.

## Results

### CVB3 infection leads to elevated levels of metabolites in the purine and pyrimidine pathways

To gain more insight into the remodeling of host metabolism by enteroviruses, a $^{13}$C-glucose isotope tracing study of cells infected with CVB3 was performed using a glucose isotopomer in which all six carbons are uniformly labeled with $^{13}$C, ([U-$^{13}$C])-glucose. Mock and infected HeLa R19 cells were supplied with medium containing [U-$^{13}$C]-glucose and subjected to metabolomics analysis at different time points after infection. The total peak intensities of metabolites in various metabolic pathways revealed that metabolic changes arise in CVB3-infected cells primarily from 4 hours post infection (hpi) (Fig 1A). Many metabolites changed upon CVB3 infection and the strongest changes were observed in purine and pyrimidine metabolism. Comparison of the total intracellular (Fig 1B) and extracellular (S1 Fig) levels of selected purines and pyrimidines in mock and CVB3-infected cells showed that the levels of nucleobases (e.g. uracil or hypoxanthine), nucleosides (e.g. uridine or guanosine), nucleotide monophosphates (e.g. GMP, UMP) and nucleotide diphosphates (e.g. GDP, UDP) strongly increased during CVB3 infection (Figs 1B and S1). In contrast, intermediates of pyrimidine *de novo* synthesis (e.g. N-carbamoyl-aspartate and dihydroorotate) strongly decreased during CVB3 infection, while their levels remained constant in mock-infected cells (Fig 1C).

### CVB3 infection increases purine and pyrimidine metabolites through nucleic acid degradation and nucleotide salvage

Three metabolic routes can contribute to an increase in purine and pyrimidine levels: 1) increased *de novo* nucleotide synthesis, 2) increased nucleic acid and nucleotide degradation, and 3) an increased activity of salvage pathways, which recycle nucleobases and nucleosides back into nucleotides. Nucleotide recycling limits the use of ATP needed for *de novo* synthesis and ensures efficient usage of available metabolites (Fig 2A and 2B). To determine which route (s) contributed to the elevation in purine and pyrimidine levels during CVB3 infection, we analysed the [U-$^{13}$C]-glucose tracing data for incorporation of $^{13}$C-carbon from glucose in nucleotide mono-, di- and triphosphates, as well as in dihydroorotate, N-carbamoyl-aspartate and orotate (Fig 2C and 2D).

Nucleotides that are synthesized *de novo* from $^{13}$C-glucose usually contain a ribose moiety that is labeled through the pentose phosphate pathway (PPP) and contributes five labeled carbons (M+5) (Fig 2A and 2B) [26]. Additionally, the carbons within the different nucleobases are labeled, albeit at a slower rate, resulting in nucleotides in which the total number of labeled carbons is either 2–4 or >5 when nucleobase labelling is combined with a labeled ribose (Fig 2B). In contrast, the degradation of preexisting nucleic acids will result in the release of unlabeled nucleotides (M+0). Finally, recycling of existing nucleobases or nucleosides into nucleotides necessitates the attachment of a (labeled) ribose moiety. Salvage therefore gives rise to M+5 labels nucleotides, but not M>5. It should be noted that although the labeling of the ribose (represented by Ribulose-5-phosphate, S2 Fig) is relatively fast with labeling approaching steady-state at 2h, labeling of the nucleobase is slower and therefore labeling of the nucleotides does not reach steady-state within our timeframe. Hence, the levels of nucleotides derived from *de novo* and/or salvage pathways are likely to be underestimated.

Examination of the labeling profiles of the nucleotides revealed that all labeled fractions (M+5, M<5 and M>5) increase over time in the mock samples, whereas in the CVB3-infected samples this increase does not persevere after 4 hpi. Instead, from 4 hpi, the unlabeled M+0 fraction remained high or even increased in CVB3-infected cells (Figs 2C and S3). Labeling of

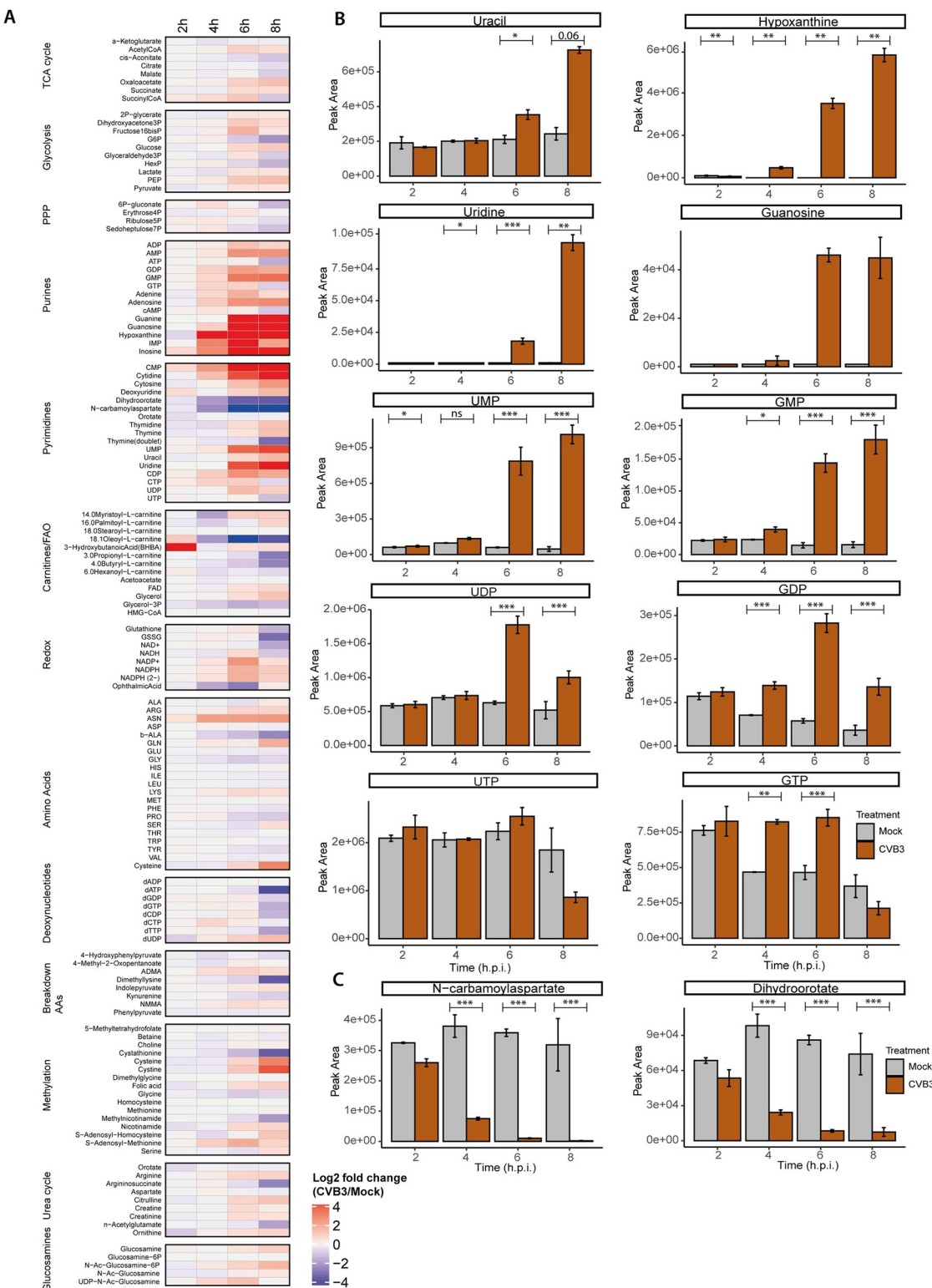

**Fig 1. CVB3 infection of triggers increases in purine and pyrimidine metabolites in HeLa cells.** [13]C-glucose isotope tracing study in mock- and CVB3-infected HeLa R19 cells (MOI 5, three replicates; one of three representative experiments is shown). Cells were infected, lysed at 2,4,6 or 8 hpi and measured by LC-MS to identify metabolites and quantify the different isotopologues. The different isotopologues are not distinguished in this Figure. A) Heatmap of log2 fold changes of the indicated metabolites between CVB3- and mock-infected cells. Log2 fold changes are calculated based on the mean of three replicates. B) Absolute peak areas of

representative nucleotide monophosphates, nucleosides and nucleobases in mock and CVB3-infected cells. C) Absolute peak areas of dihydroorotate and N-carbamoylaspartate in mock and CVB3-infected cells. The p-values were calculated using linear mixed effect models with an interaction of time and treatment and a random effect of replicate. A rank transformation on the data was performed to ensure a normal distribution of the residuals. For hypoxanthine and guanosine, a normal distribution of the residuals could not be assumed and therefore a non-parametric linear mixed effect model with an interaction of time and treatment and a random effect of replicate was performed. Afterwards, a contrast analysis was done to calculate the p-values between specific groups. $*p < 0.05$, $**p < 0.01$, $***p < 0.001$.

N-carbamoyl-aspartate, dihydroorotate and orotate followed this pattern and decreased from 4 hpi during CVB3 infection, but increased in mock-infected cells (Fig 2D). These data suggest that the relative contribution of *de novo* nucleotide synthesis to the total nucleotide pool declines during CVB3 infection, whereas that of nucleic acid degradation increases. This catabolic phenotype is further supported by the total levels of nucleotides, nucleosides and nucleobases (Figs 1A and S4). Whereas the levels of nucleotide monophosphate, nucleosides and nucleobases are increased from 6 hours onwards, the levels of nucleotide diphosphates and triphosphates decrease after 6 hours, suggesting degradation of nucleotide di- and triphosphates to nucleotide monophosphates, nucleosides and nucleobases. The third route, the salvage pathway, likely remains active in CVB3-infected cells since an M+5 fraction persists, while the M>5 fractions (almost) disappear. Also in CVB3-infected human hepatoma Huh7 cells we observed increased nucleotide levels, mainly due to an increase in the unlabeled and M+5 fractions (S5 Fig), while the M>5 fractions remain relatively stable. Furthermore, N-carbamoyl-aspartate levels decrease over time. This suggests that CVB3 infection triggers alterations in nucleotide metabolism in various cell types, albeit that the differences between cell lines warrant further study. Together, these data suggest that the rise in nucleotides and nucleotide degradation/salvage products observed during CVB3 infection is caused by nucleic acid and nucleotide degradation as well as salvage rather than through *de novo* nucleotide synthesis.

## Inhibition of nucleotide salvage constraints CVB3 replication

Our metabolomics analysis suggested that nucleotide salvage pathways remains active during CVB3 infection. To test whether these salvage pathways are also of importance for viral replication, a CVB3 luciferase reporter virus carrying a *Renilla* luciferase (Rluc CVB3) was used to determine the rate of replication in the presence and absence of salvage inhibitors. One of these inhibitors, 6-mercaptopurine (6-MP) targets HGPRT, an essential enzyme in the purine salvage pathway [27]. Indeed, 6-MP treatment of CVB3-infected HeLa R19 cells resulted in a modest but dose-dependent reduction of CVB3 replication, with the strongest effect observed at 4 hpi (Fig 3A–3C). Similar results were obtained with Cyclopentenyl uracil (CPU), an inhibitor of UCK2, the enzyme salvaging uridine and cytidine pyrimidines (Fig 3E–3G) [28]. Furthermore, a reduction of virus production was observed at 4 hpi after treating HeLa R19 cells with either 6-MP or CPU (S6 Fig). The inhibitory effects of 6-MP and CPU on replication were not due to toxic side-effects as no loss in cell viability was observed (S7 Fig). Importantly, both compounds mainly affected nucleotide metabolism and led to an increase hypoxanthine and uridine (substrates of HGPRT and UCK2, respectively) (Figs 3D, 3H and S8). This indicates that the compounds indeed inhibited the activity of these enzymes, although we cannot exclude that 6-MP has off-target effects outside of metabolism that could influence CVB3 replication. Thus, inhibiting purine and pyrimidine salvage constraints CVB3 replication especially at 4hpi. As shown above, both the absolute levels and fractions of unlabeled nucleotides rises from 4 hpi, probably due to nucleic acid degradation (Figs 2C and S4). Possibly, this rise in unlabeled nucleotides is sufficient to sustain CVB3 replication and dampens the effect of nucleotide salvage inhibition.

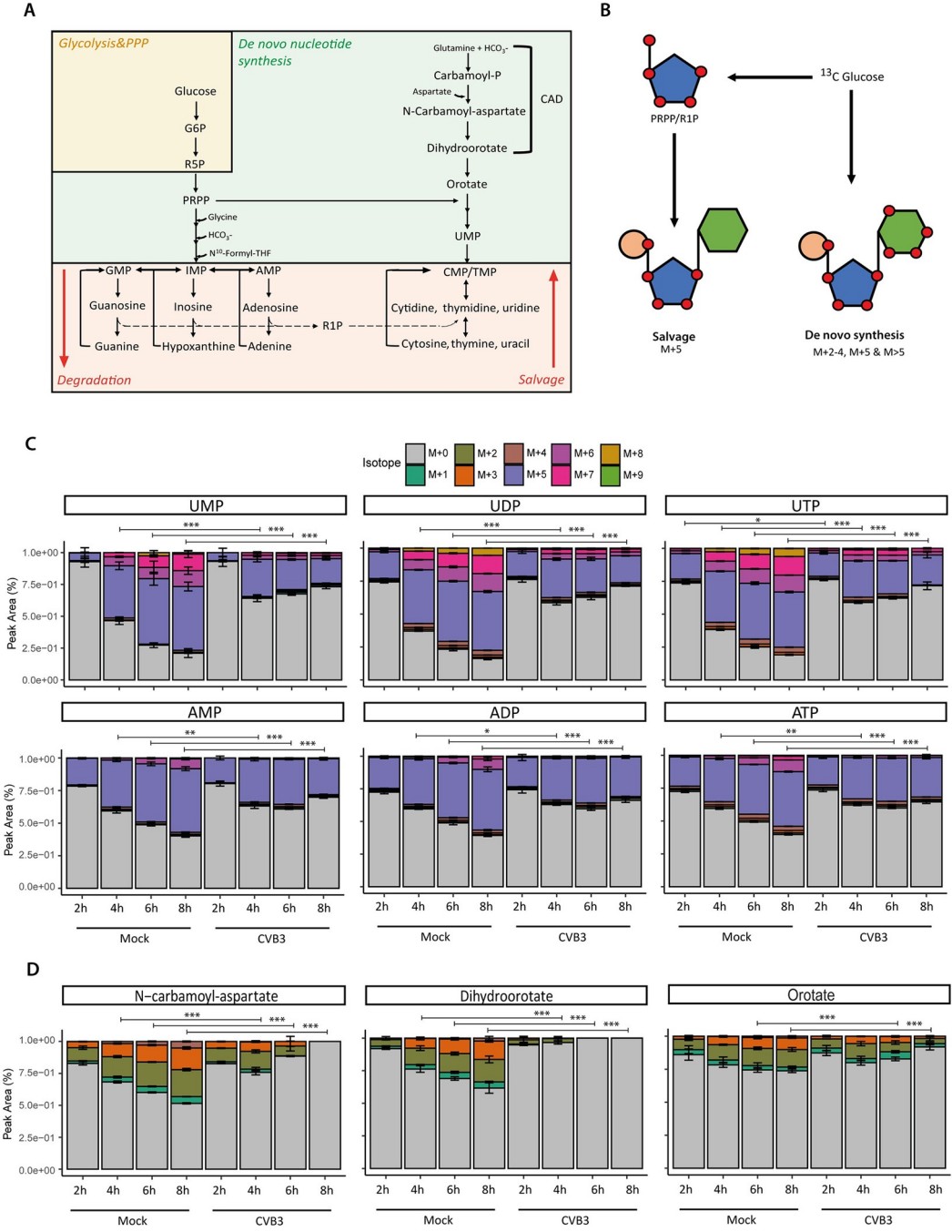

**Fig 2. CVB3 infection increases purine and pyrimidine metabolites through nucleic acid degradation and nucleotide salvage.** $^{13}$C-glucose isotope tracing study in mock- and CVB3-infected HeLa R19 cells (MOI 5, three replicates; one of three representative experiments is shown). Cells were infected, lysed at 2, 4, 6 or 8 hpi and measured by LC-MS to identify metabolites and quantify the different isotopologues. A) Schematic representation of nucleotide metabolism. Nucleotides can be synthesized *de* novo, released during degradation of nucleic acids, or recycled (i.e. called the salvage pathway). PRPP, phosphoribosyl pyrophosphate; R5P, ribose-5-phosphate; R1P, ribose-1-phosphate. B) Schematic representation of nucleotide metabolism and nucleotide labeling in $^{13}$C-glucose tracing studies. Orange = phosphate group; Blue = pentose sugar; Green = nucleobase. C) Isotope distribution of AMP, ADP, ATP, UMP, UDP and UTP in mock and CVB3-infected cells. D) Isotope distribution of dihydroorotate and N-carbamoylaspartate in mock and CVB3-infected cells. The p-values of C) and D) were calculated using linear mixed effect models with an interaction of time and treatment and a random effect of replicate. Afterwards, a contrast analysis was done to calculate the p-values between specific groups. For this analysis, the fractions of all labels were added together and tested whether the total amount of labeling differed between mock and CVB3 infection. *$p < 0.05$, **$p < 0.01$, ***$p < 0.001$.

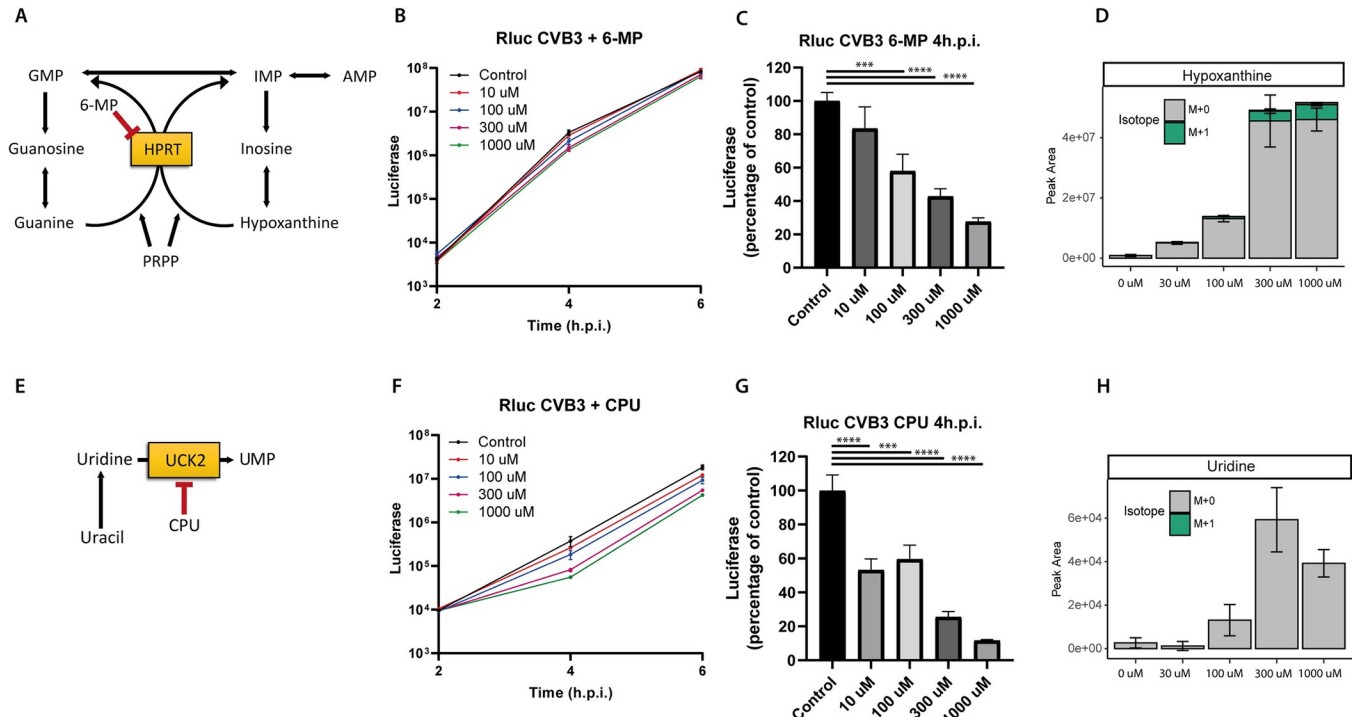

**Fig 3. Inhibition of nucleotide salvage constraints CVB3 replication.** A CVB3 luciferase reporter virus carrying a Renilla luciferase (Rluc CVB3) was used to study the replication of CVB3 in the presence or absence of compounds inhibiting the salvage pathways. A, E) Schematic representation of the purine (A) and pyrimidine (E) salvage pathway with the mode action of 6-MP (A) and CPU (E). B, F) Luciferase levels in cells infected with Rluc CVB3 (MOI 0.1) in the presence of different concentrations of 6-MP (B) or CPU (F). Cells were lysed at 2,4 or 6 hpi. Representative data of seven (B) or four (F) independent experiment are depicted (mean ± SD of 3 technical replicates). C, G) Percentage luciferase levels in 6-MP treated and CVB3-infected cells compared to control. Data represent mean ± SEM of seven (C) or four (G) independent experiments. D, H) Metabolomic analysis of hypoxanthine (D) or uridine (H) levels in 6-MP (D) or CPU (H) treated cells. Statistical analysis was done using an ANOVA. For 6-MP, not every concentration was included in all seven independent experiments, but every concentration is measured in at least three independent experiments with three replicates. ***$p < 0.001$, ****$p < 0.0001$.

## Integrative analysis of metabolomics and phosphoproteomics provides insight into the regulation of nucleotide metabolism during CVB3 infection

To deepen our understanding of how the modulation of host nucleotide metabolism during CVB3 infection is regulated, we integrated our metabolomics dataset with a previously acquired phosphoproteomic study of CVB3-infected HeLa R19 cells [29]. For this, we used MixOmics, an R package offering methods for the exploration and integration of biological datasets [30–32]. Specifically, the supervised DIABLO method was used to identify metabolites and phosphorylations that together optimally differentiate between mock and CVB3-infected cells.

A DIABLO model with two components was created (Figs 4A and S9A). Whereas the first component separates the mock samples from the CVB3-infected samples (with increasing distance over time), the second component seems to separate samples based on time, rather than infection perse. Hence, we focused on the first component for further analysis. As expected, most of the metabolites in the first component belong to nucleotide pathways (S9A Fig). The extracted phosphorylations were subjected to Gene Ontology enrichment analysis, which revealed that RNA splicing, transcription and RNA processing were among the top enriched pathways (Fig 4C).

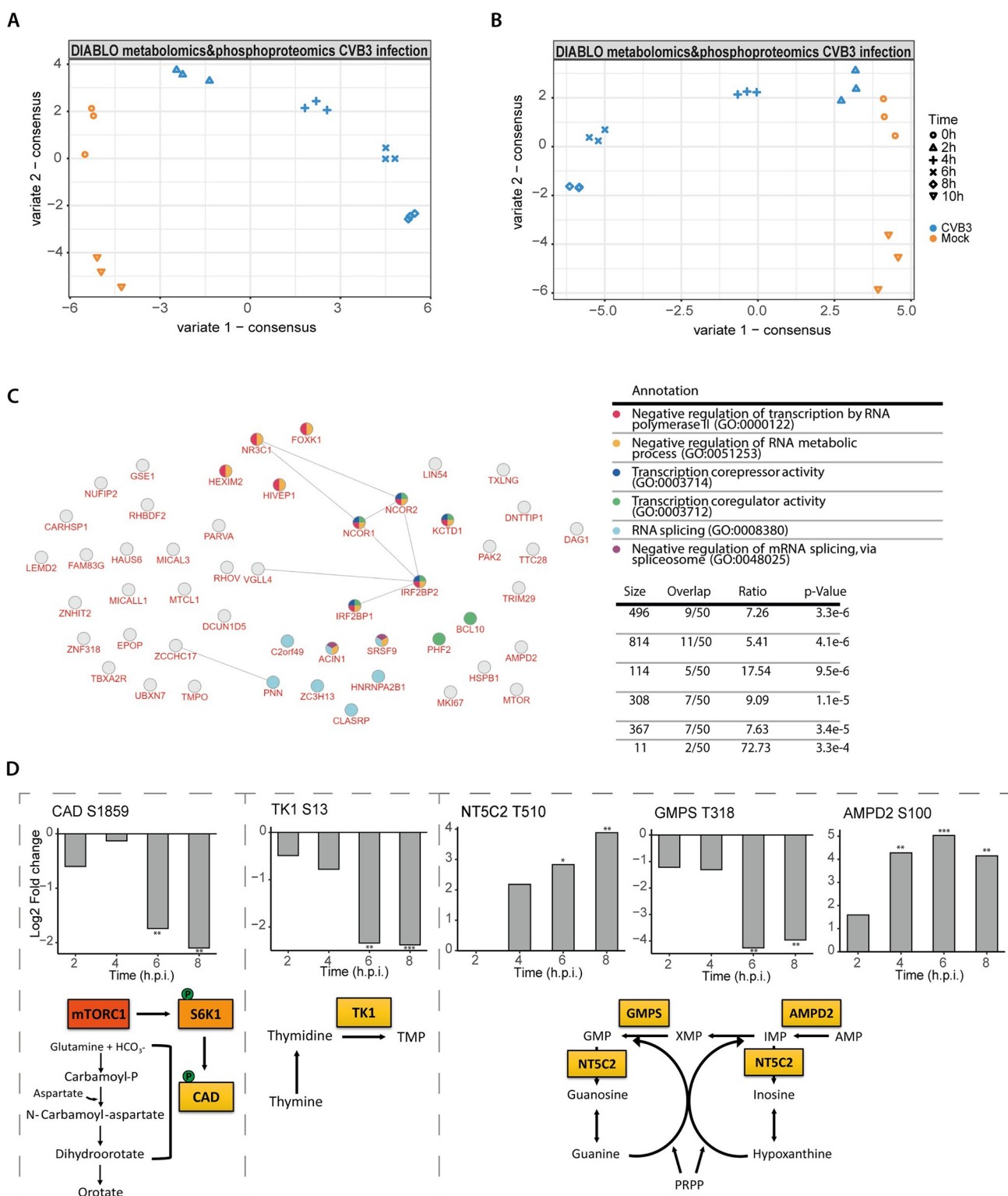

**Fig 4. Integrative analysis of metabolomics and phosphoproteomics confirms the importance of nucleotide metabolism in CVB3 infection.** A, B) DIABLO analysis (supervised analysis of multiple omics datasets) of the metabolomics and the total phosphoproteomics dataset (A) or the phosphoproteomics dataset filtered for metabolic proteins (B) of mock and CVB3-infected HeLa R19 cells. C) Enrichment analysis of proteins extracted from panel A using inBio discoverer (featuring Gene Ontology enrichment analysis). Fold enrichment of the major Gene Ontology (GO) terms is plotted, D) Regulated phosphosites on enzymes directly involved in nucleotide metabolism. Plotted are the log2 LFQ intensities of four replicates (one experiment).The p-values are calculated using ANOVA and subsequent Welch's T-tests $*p < 0.05$, $**p < 0.01$, $***p < 0.001$.

The high enrichment of RNA splicing, transcription and RNA processing pathways may mask more subtle differences in metabolic pathways. Therefore, to obtain more information on how CVB3 infection affects the phosphorylation status of metabolic proteins, the phospho-proteomic data was filtered for proteins directly involved in metabolism. This dataset was subsequently combined with the metabolomics data to create a second DIABLO model (Figs 4B and S9B). Similar to the first model, component one is primarily involved in differentiating mock samples from CVB3-infected ones and contains metabolites that mostly belong to the nucleotide pathways (S9B Fig). Five of the identified phosphosites in component one involve proteins directly involved in nucleotide metabolism, namely carbamoyl-phosphate synthetase 2, aspartate transcarbamylase, and dihydroorotase (CAD) S1859, thymidine kinase 1 (TK1) S13, AMP deaminase 2 (AMPD2) S100, GMP synthase (GMPS) T318, cytosolic purine 5'-nucleotidase (NT5C2) T510, and (Fig 4D).

Two of these phosphosites, CAD S1859 and TK1 S13, have a known function. CAD catalyzes the three initial steps in pyrimidine *de novo* nucleotide synthesis [33,34]. CAD S1859 is phosphorylated by S6K1, a direct target of mTORC1, to promote *de novo* pyrimidine synthesis [35]. mTORC1 is involved in regulating a myriad of anabolic processes and our phosphoproteomic study revealed that CVB3 infection leads to an inhibition of mTORC1 from 4–6 hpi [35]. In line with these data, CAD is dephosphorylated on S1859 during infection with CVB3 (Fig 4D), suggesting that mTORC1 inhibition contributes to inhibition of *de novo* pyrimidine synthesis via the dephosphorylation of CAD S1859. TK1 is a kinase involved in pyrimidine salvage by phosphorylating thymidine into TMP [36]. Phosphorylation of TK1 on S13 is likely mediated by CDK1 and CDK2 and inactivates this kinase (Fig 4D) [37]. During the course of CVB3 infection, TK1 S13 phosphorylation gradually decreases, suggesting that TK1 is activated and hence can contribute to nucleotide salvage.

In addition, AMPD2 S100, NT5C2 T510 and GMPS T318, phosphosites with unknown roles, were identified. AMPD2 converts AMP into IMP and is involved both in nucleotide salvage and nucleotide degradation [38]. This site is phosphorylated already early in infection and increases further over time, but the role of this phosphorylation is currently unkown. NT5C2 hydrolyses IMP and other nucleotides to nucleosides. The exact role of the phosphorylation on T510 is unknown, but this position has been implicated to regulate activation of NT5C2 [39]. Finally, GMPS, which forms GMP from XMP, is dephosphorylated on T318 from 6 hpi (Fig 4D) [40].

Enzymes in nucleotide pathways are often knocked down or out to limit cellular growth or even to trigger cell death, making genetic inhibition a less practical method to study the role of these enzyme in virus replication [41–43]. Therefore, we set out to test the effect of pharmacological inhibitors to study the role of GMPS and AMPD2 in CVB3 replication. Such inhibitors can be applied during the (relatively short) course of infection only, thereby limiting possible long-term side-effects. Both compounds affected nucleotide metabolism, especially the inhibitor of GMPS. Unexpectedly, inhibition of GMPS increased the levels of purine and pyrimidine nucleotides, including GMP levels. This increase in nucleotide levels after inhibition of GMPS was shown before in a metabolomics study of cells in which GMPS was knocked down [42]. Although inhibition of AMPD2 and GMPS affected nucleotide metabolism, we observed only a modest effect when inhibiting AMPD2 on CVB3 replication within the non-toxic concentration window (S10 Fig). This suggests that these enzymes are not essential for CVB3 replication, but does not rule out that they might have a role in fine tuning nucleotide metabolism. Together, the integrative analysis of the phosphoproteomic and metabolomic data indicates that nucleotide metabolism is affected or changing not only at the level of the metabolome, but also at the level of the phosphoproteome. Further studies are warranted to elucidate how the metabolome and phosphoproteome are regulated and intertwined, and how this influences infection.

## Metabolic changes during EMCV infection resemble those observed in CVB3 infection

We next questioned whether the observed changes in host metabolism were specific for CVB3 infection, or a more general characteristic of picornavirus infection. Therefore, we performed a side-by-side metabolomic study (without $^{13}$C-glucose tracing) comparing the metabolic changes in cells infected with CVB3 and EMCV, a picornavirus belonging to a different genus. We observed that CVB3 and EMCV induced similar responses with respect to host metabolism (r-value of 0.82) (Fig 5A and 5B). Consistently, when focusing on the metabolites in the nucleotide pathways, we found that the levels of these metabolites increased similarly, albeit stronger in cell infected with EMCV (Fig 5A). These stronger responses are likely due to a slightly higher infection rate of EMCV (S11 Fig).

To determine whether the changes in purine and pyrimidine metabolites in EMCV-infected cells are also caused by increased nucleic acid degradation and nucleotide salvage rather than *de novo* synthesis, we performed a [U-$^{13}$C]-glucose tracing study. Similar to infection with CVB3, the isotope distribution patterns of the M+5 and M>5 fractions of the nucleotides, especially for the pyrimidine nucleotides, decreased from 4 hpi, whereas the M+0 fraction increased (Fig 5C). Thus, also in EMCV-infected cells the increase in purine and pyrimidine levels are primarily caused by enhanced nucleic acid degradation and nucleotide salvage, suggesting that this mechanism is generally used for efficient picornavirus infection.

## Discussion

Viruses extensively remodel host cell functions and structures for efficient infection and spreading. Virus replication can place an increasing demand on the available cellular energy, metabolites and lipids. Therefore, many viruses alter host cell metabolism and induce glycolysis, fatty acid synthesis and/or glutaminolysis [15]. Developments in metabolic analysis techniques have significantly increased our understanding of virus-induced changes in metabolism, particularly for oncogenic viruses [44]. Here, we employed mass spectrometry-based $^{13}$C-glucose tracing methods to obtain comprehensive and detailed insight into the metabolic changes induced by picornaviruses. We present evidence that both CVB3, an enterovirus, and EMCV, a cardiovirus, increase purine and pyrimidine nucleotide levels by promoting degradation of nucleic acids and nucleotides as well as sustaining salvage pathway activity.

Our metabolomics analysis of CVB3-infected HeLa and Huh7 cells revealed a surge of nucleotides, nucleosides and nucleobases. These results are in line with a metabolomic study of rhinovirus-B14 (RV-B14) infected cells, which also observed increases in purine and pyrimidine metabolites [44]. To dissect the underlying molecular mechanism, we performed $^{13}$C-glucose tracing studies. These studies uncovered that the observed surge in CVB3-infected cells is likely caused by increased nucleic acid degradation and nucleotide salvage rather than by *de novo* nucleotide synthesis.

To explore the importance of nucleotide salvage for CVB3 replication, we inhibited salvage pathways during CVB3 infection using 6-MP and CPU. 6-MP inhibits the purine salvage pathway by blocking HRPRT, the enzyme that salvages hypoxanthine and guanine, whereas CPU inhibits the pyrimidine salvage pathway by blocking UCK2. Both inhibitors reduced the efficiency of CVB3 replication in a dose-dependent manner. This effect is strongest at 4 hpi. It is possible that the degradation of nucleic acids supplies sufficient nucleotides for CVB3 replication after this time point, reducing the effect of nucleotide salvage later in infection. These results suggest that nucleotide salvage is specifically required during the early phases of replication, but that this need declines later in infection due to the availability of nucleotides derived from nucleic acid degradation.

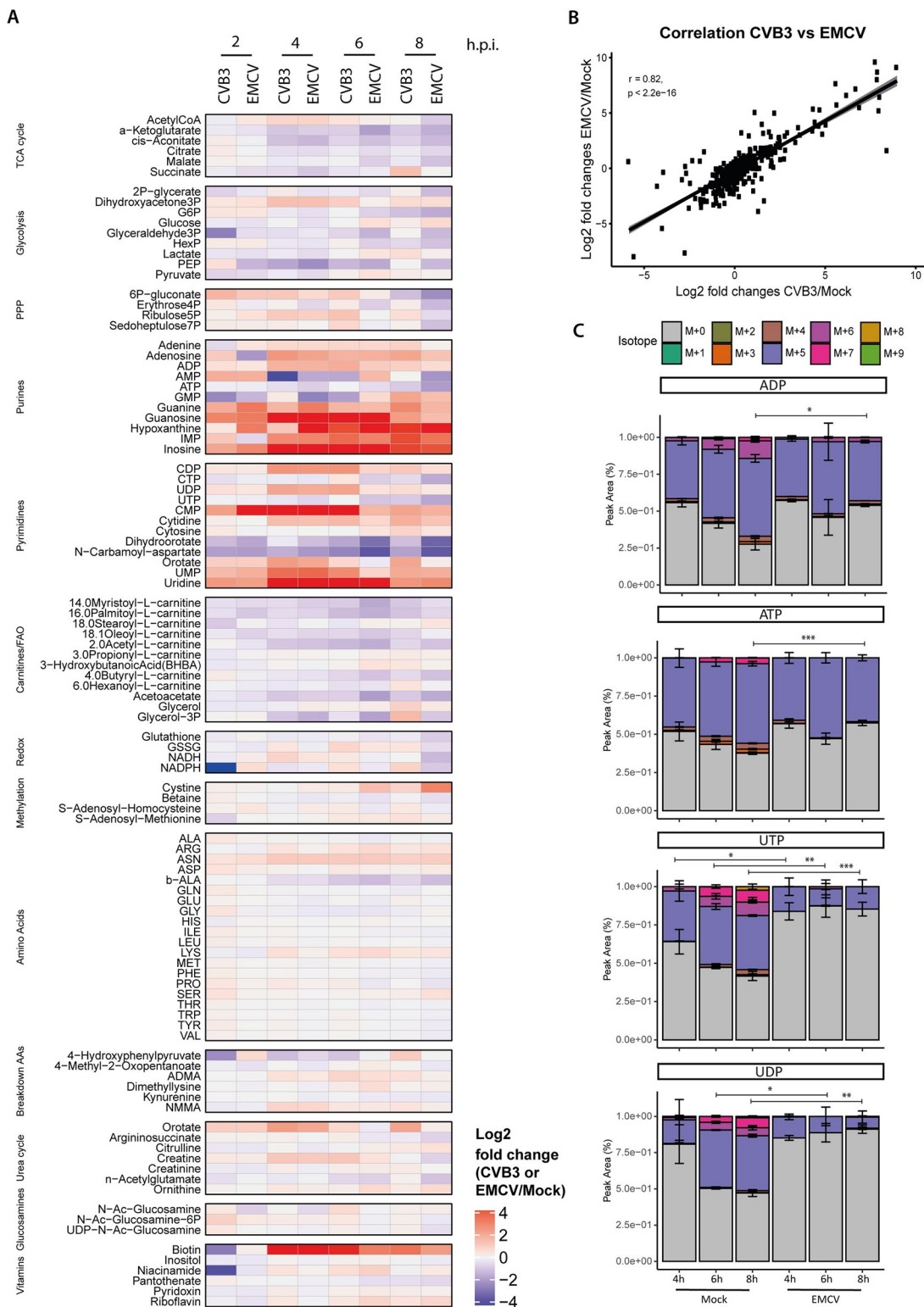

**Fig 5. Metabolic changes during EMCV infection resemble those observed in CVB3 infection.** A) Metabolomics study of mock- and CVB3- or EMCV-infected HeLa R19 cells (MOI 5). Cells were lysed at 2, 4, 6 or 8 hpi and subjected to LC-MS for the measurement of metabolites. Heatmap showing log2 fold changes in metabolite levels in CVB3- and EMCV-infected cells compared to mock infection (three replicates; one experiment for each virus). One sample was removed from analysis (one replicate of 2 hpi CVB3), because of a technical defect. B) A Pearson correlation test was performed to determine the level of

correlation between metabolic changes induced by CVB3 and EMCV. C) $^{13}$C-glucose isotope tracing study in mock- and EMCV-infected HeLa R19 cells (three replicates; one experiment; MOI 5). Cells were infected, lysed at 2, 4, 6 or 8 hpi and measured by LC-MS to identify metabolites and quantify the different isotopologues. Isotope patterns in ADP, ATP, UDP and UTP in mock and EMCV infected HeLa R19 cells are depicted. One sample was removed from analysis (one replicate of 8 hpi EMCV), because of a technical defect. The p-values were calculated using linear mixed effect models with an interaction of time and treatment and a random effect of replicate. A rank transformation on the data was performed to ensure a normal distribution of the residuals. Afterwards, a contrast analysis was done to calculate the p-values between specific groups. For this analysis, the fractions of all labels were added together and tested whether the total amount of labeling differed between mock and CVB3 infection. $^{*}p < 0.05$, $^{**}p < 0.01$, $^{***}p < 0.001$.

To better understand what drives the changes in nucleotide metabolism, we integrated the metabolomic data with those from our previously described phosphoproteomic analysis of CVB3-infected HeLa cells, filtered for metabolic enzymes. This integration resulted in the identification of five phosphorylations that significantly change during viral infection on cellular enzymes directly involved in nucleotide metabolism. Three of these enzymes function in nucleotide degradation and/or nucleotide salvage. We observed dephosphorylation of TK1 on S13, known to result in activation of TK1, in infected cells [45–47]. TK1 phosphorylates thymidine into TMP and is thereby involved in salvaging nucleotides. This observation supports the idea that pyrimidine salvage is enhanced during CVB3 infection. We also observed phosphorylation of NT5C2, specifically on T510, a position that has been implicated in regulating NT5C2 activity. NT5C2 hydrolyzes IMP and other nucleotides to nucleosides and is therefore involved in nucleotide degradation. Notably, the analysis also retrieved the phosphorylation of AMPD2 on S100. Although the role of this particular phosphorylation is currently unknown, it is well established that AMPD2 converts AMP to IMP. Because IMP can be converted to both AMP and GMP, AMPD2 bridges degradation and salvage of nucleotides and in doing so regulates the balance of GMP and AMP levels. However, inhibiting AMPD2 only modestly affected CVB3 replication, suggesting that the activity of this enzyme is not essential for CVB3 infection. Although not essential for CVB3 replication, AMPD2 could still play a role in balancing AMP and GMP levels, also depending on the role of the phosphorylation of S100 on AMPD2. Collectively, the phosphorylations highlighted by the integrative analysis suggest that nucleotide degradation and nucleotide salvage during CVB3 infection are not only altered at the metabolome, but also on the phosphoproteome levels. Further studies are warranted to establish the role of the phosphorylations we find during CVB3 infection.

The regulation of nucleic acid metabolism in CVB3 infection, in particular RNA metabolism, was underpinned by the integrative analysis of the metabolomic data with the phosphoproteomic data. Upon extracting phosphorylations that differentiated mock from infected samples, phosphorylations on proteins involved in the regulation of RNA splicing, transcription and RNA metabolism were enriched. Since RNA splicing and RNA degradation are tightly connected, it is possible that alterations in splicing and mRNA metabolism may underly RNA degradation [45–47]. Enteroviruses modulate splicing through several mechanisms. By disrupting nucleocytoplasmic trafficking, these viruses alter the location of splicing factors and disturb the splicing machinery within the host cell [48]. Moreover, 3Dpol, the enteroviral RNA-dependent RNA polymerase, enters the nucleus and targets the pre-mRNA processing factor 8 (Prp8) to block pre-mRNA splicing and mRNA synthesis [49]. Additionally, enteroviruses inhibit cap-dependent host mRNA translation by cleaving the translation initiation factor eIF4G. In non-infected cells, inhibition of translation induces the formation of stress granules (SGs) to protect the mRNAs from degradation until translation is resumed. These SGs are functionally and physically connected to processing bodies (PBs), another type of cytoplasmic RNA granule that is involved in the decay of non-translating mRNAs. During infection, enteroviruses trigger the formation of SGs early in infection wherease they disassemble

SGs and PBs and inhibit their formation later in infection. These dynamic effects likely alter the affect the balance of cellular mRNA stalling and degradation [50,51]. Thus, the enteroviral deregulation of splicing, translation and SG formation may lead to increased RNA degradation, thereby contributing to the elevated levels of nucleotides, nucleosides and nucleobases in infected cells.

Aside from SGs and PBs, picornaviruses also have an intricate interplay with other RNA degradation pathways. In general, the literature is focused on how viruses, particularly RNA viruses, evade RNA decay pathways to protect their viral genome. One of these pathways is the RNAse L pathway that is activated when dsRNA intermediates are formed and sensed by the host innate defense mechanism. Upon activation of RNase L both viral and cellular RNA get degraded. Picornaviruses have evolved mechanisms to suppress the activation of RNase L either directly or indirectly by inhibiting the sensing or activation of innate immune pathways [50,51]. In addition, two important proteins involved in RNA degradation, XRN1, the major $5' \rightarrow 3'$ exonuclease, and AUF1, an important ARE-binding protein that promotes the decay of numerous mRNAs, are cleaved during infection via the proteasome or the viral protease 3C[pro], respectively, to reduce RNA degradation [52–55]. Notably, this inhibition is likely incomplete as depletion of XRN1 or AUF1 increases enteroviral RNA replication and virus titers [53–56]. Moreover, autophagy, which is tightly conntected to RNA decay pathways and which is activated during enterovirus infection, also likely contributes to the observed RNA degradation and the subsequent release of nucleotides [57–59]. Apart from antiviral effects of RNA degradation, we provide evidence that the degradation of nucleic acids increases nucleotide levels, which might also be proviral. However, how antiviral and proviral RNA degradation are regulated during CVB3 infection remains to be established.

Apart from increased nucleic acid and nucleotide degradation, the relative contribution of *de novo* synthesis to the total nucleotide levels was decreased in CVB3 infection. Our integrative analysis emphasized the importance of phosphorylation of CAD on S1859 during viral infection. This site is specifically phosphorylated by mTORC1 and activates CAD. In line with mTORC1 inhibition, a decrease of the phosphorylation on CAD S1859 was observed, suggesting that inhibition of mTORC1 inhibits CAD activity, resulting in the restriction of *de novo* pyrimidine synthesis [35]. Also, another study revealed that FMDV inhibits Histone deacetylase 1 (HDAC1) and subsequently mTORC1 and CAD during infection [60]. Elevated levels of nucleotides have also been described in Vero cells infected with EV-A71 [28]. In that study, however, no inhibition of CAD was observed. Instead, increased CAD activity was described, suggestively due to an interaction between viral capsid protein VP1 and CAD. Whether these results reflect fundamental differences between picornaviruses, cell lines or experiment conditions remains to be established.

How enteroviruses interfere with *de novo* purine synthesis remains unknown, but several, not mutually exclusive, mechanisms may be involved. Our integrative analysis points to virus-induced alterations in the phosphorylation status of GMPS at T318, the enzyme catalyzing the generation of GMP from XMP, hinting to viral effects of GMP synthesis. Whether this phosphorylation activates or inhibits GMPS is currently unknown, but pharmacological inhibition of this enzyme did not affect CVB3 replication, suggesting GMPS activity is not essential for CVB3 replication. Apart from this, 3C[pro] of CVB3 and poliovirus have been described to cleave phosphoribosylformylglycinamidine synthase (PFAS), an enzyme involved in *de novo* nucleotide purine synthesis [61]. Notably, PFAS also plays a role in RIG-I activation. Herpesviruses target PFAS to modulate RIG-I activation and the localization of the transcription factor replication transactivator (RTA) that is crucial for KSHV lytic replication [62,63]. Although it is well established that enteroviruses and other picornaviruses are predominantly sensed by MDA5, cell type specificity and functional redundancy of MDA5 and RIG-I has been

described [64]. Growing evidence indicates that nucleotide metabolism, particularly pyrimidine metabolism, is thightly linked to innate immune responses [65–68]. The link between nucleotides and innate immunity makes nucleotide metabolism attractive for a virus to interfere with. Whether enteroviruses target pyrimidine and purine *de novo* synthesis to modulate nucleotide metabolism, to modify other cellular pathways (e.g. RIG-I signaling), or both, requires further exploration.

To determine whether other picornaviruses modulate host metabolism similarly to CVB3, we also performed a metabolomic analysis of cells infected with EMCV, a cardiovirus. Similar to CVB3, EMCV infection increased the levels of nucleotides, nucleosides and nucleobases. EMCV infection decreased levels of two intermediates of the pyrimidine synthesis pathway, dihydroorotate and N-carbamoyl-aspartate, to a similar extent as CVB3. Although this is suggestive to conserved mechanism, subtle differences in the regulation of mTORC1 and hence *de novo* nucleotide synthesis during infection with CVB3 and EMCV have been observed [28,68–70].

In conclusion, we here demonstrate that the picornaviruses CVB3 and EMCV extensively reprogram nucleotide metabolism and we present evidence that the alterations in nucleotide metabolism are likely due to the activation of nucleic acid degradation and salvage pathways. Our data are important for better understanding of the virus-host interaction and may lead to new avenues for therapeutic intervention.

## Materials & methods

### Cells and viruses

Hela R19, HEK293T and Huh7 cells were maintained in Dulbecco's modified Eagle's medium (DMEM; Lonza) supplemented with 10% fetal bovine serum (FBS) and 1% Pen-Strep (Lonza). BHK21 cells were cultured in DMEM containing sodium pyruvate and glutamax (Gibco; 2206106) and supplemented with 10% FBS and 1% Pen-Strep (Lonza).

CVB3 was generated by passaging the virus on Hela R19 cells. Upon complete CPE, the virus was harvested and concentrated by ultracentrifugation (30% sucrose, 140.000 g for 16 hours, 4°C, SW32Ti rotor). The virus was subsequently diluted in PBS and stored at -80C. EMCV and *Renilla* luciferase (Rluc)-CVB3 viruses were generated by producing RNA from their respective infectious clones and transfecting this RNA into BHK21 or HEK293T cells respectively. Similar to the production of CVB3, viruses were harvested after complete CPE, concentrated by ultracentrifugation, diluted in PBS and stored at -80°C. Virus titers were determined by end point titration on Hela R19 and BHK21 cells according to the method of Spearman-Kärber and expressed as 50% Tissue Culture Infectious dose (TCID50).

### Effects of compounds on CVB3 replication

To study the effect of compounds on CVB3 replication, we used a *Renilla* luciferase (RLuc)-expressing CVB3 [69]. Briefly, HeLa R19 cells ($1*10^4$ cells/well) were seeded in 96 well plates. The following day, the cells were infected with *Renilla* luciferase (RLuc)-CVB3 for 30 minutes at a multiplicity of infection (MOI) of 0.1. after which the inoculum was removed and fresh (compound-containing) medium was added to the cells. The cells were lysed with 50μl lysis buffer (Promega) at the appropriate time points and stored at -20°C for further use. In order to measure the luminescence of the cell lysates, a *Renilla* assay system (Promega) was used according to manufacturer's protocol. For analysis of the luciferase assays Graphpad Prism (version 8) was used.

To measure the effect of compounds on cell viability, the CellTiter 96 AQueous solution One Solution Cell Proliferation Assay (MTS) was used (Promega). Hela R19 cells were seeded

in 96 wells plates at a density of $1*10^4$ cells/well. After incubation with the compound for 6 or 24 hours, the medium was refreshed and MTS was added (10 μl MTS/50 μl medium). This mix was incubated for 1–2 hours at 37°C after which the absorbance was measured with an Elisa Plate reader at 490 nm. Viability was determined as the percentage viable cells compared to the control (cells without compound).

### Synthesis Cyclopentenyl uracil (CPU)

CPU was synthesized according to known literature procedures. The spectral data was in accordance with that of literary precedence [28,70–72].

### Metabolite profiling and isotope tracing

For the metabolomic studies, HeLa R19 or Huh7 cells were seeded in 6 well plates at a density of $4*10^5$ cells/well or $5*10^5$ cells/well respectively. The next day, the cells were infected with the corresponding viruses (diluted in DMEM with 10% FBS) for 30 minutes. After 30 minutes, the medium was refreshed with either DMEM with 10% FBS for regular metabolomics studies or DMEM supplemented with 10%FBS, 2mM glutamine and either 25mM [U-$^{13}$C]glucose (Cambridge Isotopes) or $^{12}$C-Glucose for isotope tracing studies. The cells were subsequently incubated for the given time points. When a time point was reached, cells were washed with PBS and lysed with 1 ml of lysis buffer consisting of methanol/acetonitrile/H2O (2:2:1) for metabolite extraction. Cell lysates were centrifuged at 15.000 g for 15 minutes (4°C) and supernatant was collected for liquid chromatography mass spectrometry (LC-MS) analysis.

The samples in Figs 1, 2 and S1–S4 were analysed on an Exactive mass spectrometer (Thermo Scientific) coupled with a Dionex Ultimate 3000 autosampler and pump (Thermo Scientific). The samples depicted in Figs 5, S5, S8 and S10 were run on a Q-Excative HF Quadrupole-orbitrap mass spectrometer (Thermo Scientific). Both machines operated in polarity-switching mode with spray voltages of 4.5 kV and -3.5 kV. Metabolites within all samples were separated using a Sequant ZIC-pHILIC column (2.1 × 150 mm, 5 μm, guard column 2.1 × 20 mm, 5 μm; Merck). The solvents used are acetonitrile and eluent A (20 mM (NH4)2CO3, 0.1% NH4OH in ULC/MS grade water; Biosolve). The flow rate was set on 150 ul/min with a gradient of 20–60% of eluent A in 20 minutes. After 20 minutes, the column was washed with 80% A and requalibrated with 20% A.

After sample acquisition, metabolites were identified and quantified using either LCquan software (Thermo Scientific; Figs 1, 2 and S1–S4) or TraceFinder software (Thermo Scientific; Figs 5, S5, S8 and S10) both on the basis of the exact mass within 5 ppm. Additionally, the retention times of a set of standards were used to further validate the identification of the metabolites. Peak intensities were normalized using mean peak intensities of total metabolites. Isotope distributions were corrected for natural abundance of $^{13}$C.

### Statistics

The metabolomic data was analyzed using R. Heatmaps were generated using the Complex-Heatmap R package [73]. To statistically analyze the metabolomic data, a linear mixed effect model with an interaction of time and treatment and a random effect of replicate was performed (using the lme4 R package [74]). The normal distribution of the residuals was checked per metabolite and if needed a rank transformation was performed. In case normality could still not be assumed, a non-parametric linear mixed effect model with an interaction of time and treatment and a random effect of replicate was performed (using the ARTool R package [75]). After creating the appropriate model, the model was used to perform contrast analyses to test the differences between specific groups (using the emmeans R package [76]). Multiple

testing correction was performed using the Benjamin-Hochberg method. The p-values for the phosphoproteins where taken from the previously published and publicaly available phospho-proteomic dataset and analysis [29]. Other statistical analysis were performed using Graphpad Prism 9 (exact method specified in the dedicated figure legend).

### Integrative omics analysis

The metabolomic and phosphoproteomic data was integrated using the R package MixOmics [31]. To use MixOmics, both datasets should contain the same number of groups and replicates per group. The phosphoproteomic datasets only contained 0h mock and 10h mock samples and 4 replicates per time point, whereas the metabolomic screen included mocks at all time points (0, 2, 4, 6, 8 and 10hpi) and 3 replicates per time point. Therefore, only the 0h mock and 10h mock samples were included in the integrative analysis, and 3 out of 4 replicates from the phosphoproteomics experiment were randomly sampled. The DIABLO analysis requires the user to specify the number components to create and the number of variables to be selected per component per dataset. Guided by the MixOmics tunig functions the following constrainst were chosen: 2 components; component 1: 15 metabolites, 55 phosphosites; component 2: 10 metabolites, 25 phosphosites. Pathway/enrichment analysis of metabolomic data was performed with MetaboAnalyst, while for the enrichment analysis of the phosphoproteomic in Bio discoverer (featuring Gene Ontology enrichment analysis) was used.

### Supporting information

**S1 Fig. Metabolomic analysis of the medium of CVB3 infected Hela cells.** $^{13}$C-glucose isotope tracing study in mock- and CVB3-infected HeLa R19 cells (three replicates; one experiment; MOI 5). Cells were infected, lysed at 2,4,6 or 8 hpi and measured by LC-MS to identify metabolites and quantify the different isotopologues. The different isotopologues are not distinguished in this Figure. Heatmap showing log2 fold changes of extracellular metabolites during CVB3 infection using.
(TIF)

**S2 Fig. Isotope distribution of Ribulose-5-Phosphate.** $^{13}$C-glucose isotope tracing study in mock HeLa R19 cells (three replicates; one experiment; multiplicity of infection (MOI) = 5). Cells were lysed at 2,4,6 or 8 hpi and measured by LC-MS to identify metabolites and quantify the different isotopologues. Ribulose-5-Phosphate is depicted.
(TIF)

**S3 Fig. Isotope distribution of guanine and cytosine nucleotides in CVB3 infected Hela R19 cells.** $^{13}$C-glucose isotope tracing study in mock- and CVB3-infected HeLa R19 cells (three replicates; one experiment; multiplicity of infection (MOI) = 5). Cells were infected, lysed at 2,4,6 or 8 hpi and measured by LC-MS to identify metabolites and quantify the different isotopologues. In one replicate of 6h mock, CMP was undetectable leading to inaccurate fraction calculations. Therefore, the 6h mock was omitted in the CMP figure. The p-values were calculated using linear mixed effect models with an interaction of time and treatment and a random effect of replicate. For CMP, a normal distribution of the residuals could not be assumed and therefore a non-parametric linear mixed effect model with an interaction of time and treatment and a random effect of replicate was performed. Afterwards, a contrast analysis was done to calculate the p-values between specific groups. For this analysis, the fractions of all labels were added together and tested whether the total amount of labeling differed between mock and CVB3 infection. $^{*}$p $< 0.05$, $^{**}$p $< 0.01$, $^{***}$p $< 0.001$.
(TIF)

**S4 Fig. Absolute nucleotide levels of metabolomic screen of CVB3 infected Hela R19 cells.**
Metabolomics study of mock and CVB3 infected HeLa R19 cells (three replicates; one experiment; MOI 5). The cells were infected, lysed at 2, 4, 6 or 8 hpi and measured by LC-MS to identify metabolites. The absolute levels of nucleotide mono-, di- and triphosphates are shown. The p-values of the absolute metabolite levels were calculated using linear mixed effect models with an interaction of time and treatment and a random effect of replicate. A rank transformation on the data was performed to ensure a normal distribution of the residuals. Afterwards, a contrast analysis was done to calculate the p-values between specific groups. $*p < 0.05$, $**p < 0.01$, $***p < 0.001$.
(TIF)

**S5 Fig. Nucleotide changes in CVB3 infected Huh7 cells.** [13]C glucose isotope tracing study of mock and CVB3 infected Huh7 R19 cells (three replicates; one experiment; MOI 5). The cells were infected, lysed at 5,8 or 16 hpi and measured by LC-MS to identify metabolites. One sample was removed from analysis (one replicate of 5 hpi CVB3), because of a technical defect. A) The relative contributions of the different labelings are shown for eight representative nucleotides: ADP, ATP, UDP, UTP, CDP, CTP, GDP, GTP. B) The absolute level of N-carbanoyl-aspartate in Mock and CVB3 infected samples over time. The p-values of the absolute metabolite levels A) and B) were calculated using linear mixed effect models with an interaction of time and treatment and a random effect of replicate. For N-carbamoyl-aspartate, a normal distribution of the residuals could not be assumed and therefore a non-parametric linear mixed effect model with an interaction of time and treatment and a random effect of replicate was performed. Afterwards, a contrast analysis was done to calculate the p-values between specific groups. $*p < 0.05$, $**p < 0.01$, $***p < 0.001$.
(TIF)

**S6 Fig. Inhibition of nucleotide salvage constraints CVB3 virus production early in infection.** Growth kinetics of CVB3 in HeLa R19 cells in the presence of different concentrations of 6-MP (A, B) or CPU (C, D). The cells were infected, treated with the compounds directly after infection, lysed at 2, 4, 6, or 10 hpi and titrated on HeLa R19 cells to determine the TCID50/ml (MOI 5; mean ± SEM of triplicates; one experiment). Guanidine hydrochloride (GuaHCl) is a known replication inhibitor. A,C) Growth curve of all the included time points. B,D) Bar graph of the 4h time point. A two-way ANOVA was performed, but results are not significant.
(TIF)

**S7 Fig. Viability measurements after 6-MP or CPU treatment.** A) A representative MTS assay performed in parallel with the luciferase assay depicted in Fig 3A (mean ± SD). The cells were exposed to the different 6-MP concentrations for either 6h or 24h after which a MTS assay was used to determine the viability of the cells. B) A representative MTS assay performed in parallel with the luciferase assay depicted in Fig 3E (mean ± SD). The cells were exposed to the different CPU concentrations for either 6h or 24h after which a MTS assay was used to determine the viability of the cells.
(TIF)

**S8 Fig. Metabolomic analysis of 6-MP and CPU treated cells.** [13]C-glucose isotope tracing study in control- and 6-MP or CPU treated HeLa R19 cells (three replicates; one experiment). Cells were exposed to different concentrations of 6-MP or CPU (30 μM, 100 μM, 300 μM, 1000 μM). The cells were lysed after at 6h and measured by LC-MS to identify metabolites and quantify the different isotopologues. The different isotopologues are not distinguished in this Figure. Heatmap showing log2 fold changes of intracellular metabolites compared to control-

treated cells.
(TIF)

**S9 Fig. The extracted metabolites and phosphorylations from component one from the integrated DIABLO analysis.** A) Extracted metabolites and phosphorylations of the DIABLO analysis (supervised analysis of multiple omics datasets) of the metabolomics dataset (MOI 5) and the total phosphoproteomic dataset (MOI 10) of mock and CVB3 infected HeLa R19 cells. The phosphorylations contain the protein, the phosphorylated site and whether this site is quantified on a singly (_1) or doubly (_2) phosphorylated peptide. B) Extracted metabolites and phosphorylations of the DIABLO analysis (supervised analysis of multiple omics datasets) of the metabolomics dataset (MOI 5) and the phosphoproteomic dataset filtered for metabolic proteins (MOI 10) of mock and CVB3 infected HeLa R19 cells. The phosphorylations contain the protein, the phosphorylated site and whether this site is quantified on a singly (_1) or doubly (_2) phosphorylated peptide.
(TIF)

**S10 Fig. The effect of inhibition of AMPD2 and GMPS on cellular metabolism and CVB3 replication.** A CVB3 luciferase reporter virus carrying a *Renilla luciferase* (Rluc CVB3) was used to study the replication of CVB3 in the presence or absence of compounds inhibiting the salvage pathways. A, C) Luciferase levels in cells infected with Rluc CVB3 (MOI 0.1) in the presence of different concentrations of Decoyinine (A) or AMPD2 inhibitor (C). Cells were lysed at 2, 4, 6, 8 hpi. Representative data of three independent experiment are depicted (mean ± SD of 3 technical replicates). B,D) MTS assay performed in parallel with the luciferase assay depicted in A and C (mean ± SD). The cells were exposed to the different Decoyinine (B) or AMPD2 inhibitor (D) concentrations for either 8h after which a MTS assay was used to determine the viability of the cells. E) $^{13}$C-glucose isotope tracing study in control- and decoyinine (100 μM) or AMPD2 inhibitor (100 μM) treated HeLa R19 cells (three replicates; one independent experiment). The cells were lysed after 6h and measured by LC-MS to identify metabolites and quantify the different isotopologues. The different isotopologues are not distinguished in this Figure. Heatmap showing log2 fold changes of intracellular metabolites compared to control-treated cells.
(TIF)

**S11 Fig. Growth kinetics CVB3 and EMCV in HeLa R19 cells.** Growth kinetics of CVB3 and EMCV in HeLa R19 cells, titrated on HeLa R19 cells (MOI 2; mean and SD of triplicates; one experiment). The cells were infected, lysed at 2, 4, 6, 8 or 10 hpi and titrated to determine the TCID50/ml.
(TIF)

**S1 Data. Datasets.**
(ZIP)

## Acknowledgments

We thank Erik de Vries for his constructive feedback on this manuscript.

## Author Contributions

**Conceptualization:** Lonneke V. Nouwen, Esther A. Zaal, Celia R. Berkers, Frank J. M. van Kuppeveld.

**Formal analysis:** Lonneke V. Nouwen, Chris H. A. van de Lest.

**Investigation:** Lonneke V. Nouwen, Martijn Breeuwsma, Inge Buitendijk, Marleen Zwaagstra.

**Resources:** Pascal Balić, Dmitri V. Filippov, Celia R. Berkers.

**Supervision:** Esther A. Zaal, Celia R. Berkers, Frank J. M. van Kuppeveld.

**Visualization:** Lonneke V. Nouwen.

**Writing – original draft:** Lonneke V. Nouwen.

**Writing – review & editing:** Esther A. Zaal, Celia R. Berkers, Frank J. M. van Kuppeveld.

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
