## [Decision Letter · Decision Letter 0]

13 Nov 2023

Dear  Frank,

Thank you very much for submitting your manuscript "Modulation of nucleotide metabolism by picornaviruses" for consideration at PLOS Pathogens. As with all papers reviewed by the journal, your manuscript was reviewed by members of the editorial board and by several independent reviewers. The reviewers appreciated the attention to an important topic. Based on the reviews, we are likely to accept this manuscript for publication, providing that you modify the manuscript according to the review recommendations.

In the reviesed manuscript, please avoid the overinterpretation of the resutls, this was a concern of all reviewers, and confirm the key results using infectious virus system, not only the luciferase replicon

Sincerely,

George A. Belov, PhD

Academic Editor

PLOS Pathogens

Guangxiang Luo

Section Editor

PLOS Pathogens

Kasturi Haldar

Editor-in-Chief

PLOS Pathogens

orcid.org/0000-0001-5065-158X

Michael Malim

Editor-in-Chief

PLOS Pathogens

orcid.org/0000-0002-7699-2064

Reviewer Comments (if any, and for reference):

Reviewer's Responses to Questions

**Part I - Summary**

Reviewer #1: The authors conducted tracing experiments using [13C]-glucose to investigate the metabolome of infected cells at various time points throughout the infection. This approach allowed for a more accurate identification of the activities within metabolic pathways, which in turn provided a thorough comprehension of how the virus affects metabolic regulation. Additionally, the authors utilized previously acquired phosphoproteomic data from CVB3-infected cells to gain further insight into the underlying mechanisms that drive metabolic alterations during infection. The findings of this study provide compelling evidence that both CVB3 and EMCV infections enhance metabolite levels in the purine and pyrimidine pathways by promoting nucleic acid and nucleotide degradation, as well as facilitating nucleotide recycling, rather than through increased de novo synthesis of nucleotides.

This study is relevant because our understanding of how picornaviruses remodel cellular metabolism remains limited. Generally, viruses have the capability to actively reprogram glucose, glutamine, fatty acids, and nucleotides metabolism. This reprogramming not only facilitates the availability of necessary building blocks for viral replication but also acts in response to innate immune responses.

Interestingly, when a cell is infected with CVB3 at 4 hpi, the levels of nucleobases (such as uracil or hypoxanthine), nucleosides (such as uridine or guanosine), nucleotide monophosphates (like GMP, UMP), and nucleotide diphosphates (such as GDP, UDP) experience a significant increase. In contrast, intermediates of pyrimidine de novo synthesis show a pronounced increase during CVB3 91 infection. It would be good to provide folds of change to better represent the changes in nucleotide accumulation.

Reviewer #2: This manuscript reports the effect of picornavirus infection on nucleotide metabolism and its role in viral replication. The study is primarily based on carbon flux metabolomics and phosphoproteomics to make correlations between signal transduction via mTOR and changes in nucleotide metabolism in cells.

Reviewer #3: In this study the authors use a HELA cell line to study metabolic changes induced by picornaviruses, CVB3 and in one experiment, EMCV. For CVB3 they show metabolomic data indicating that infection increases pools of nucleobases, nucleosides, NMPs and NDPs but interestingly not NTPs. They also find strong decreases in pyrimidine synthesis intermediates which indicates that de novo nucleotide synthesis may be inhibited. They also use uniformly labeled C-13 glucose tracing to show that after infection there is decreased higher level labeling (M >5) of nucleosides, NMPs and NDPs further indicating that there is likely inhibition of de novo purine and pyrimidine synthesis. They show that the levels of (M + 5) typically derived from the labeled ribose ring and the nucleotide salvage pathway do not strongly increase after 4 hours as it does in the uninfected cells. However, they claim multiple times that this indicates increased salvage pathway, but the data do not clearly show this. While it is possible due to increased overall purine and pyrimidine levels that the salvage pathway increases somewhat, they do not clearly show this. They also perform metabolomic and C-13 glucose tracing in HUH cells, a transformed liver line and see many similar results. However, they do not address that there is a major difference between cell lines and that there is an increase in NTPs as well as NMPs and NDPs. Drug inhibition of the purine or pyrimidine salvage pathway leads to modest inhibition of viral replication, though they don’t show production of infectious virus, only luciferase assays. They never explain the decrease in NTPs and the relationship to viral replication, though this could be relevant. They include earlier phosphor-proteomic data to show that a number of nucleotide synthesis and degradation pathway enzymes are differentially phosphorylated. However, none of these studies are backed up with viral replication data or significant evidence they are invoved. Finally they show metabolomic data for ECMV and find many similarities in purine and pyrimidine metabolism to CVB3 infected HELA cells.

**Part II – Major Issues: Key Experiments Required for Acceptance**

Reviewer #1: Given that [13C]-glucose is introduced to the culture simultaneously with the initiation of infection, one important point to clarify is the speed at which the incorporation of [13C] from de novo synthesis of nucleotides is expected to occur. This is particularly crucial considering the rapid replication of the virus. Are there any positive controls, especially during the early timepoints, to determine if labeling has not yet reached a steady-state? This would help assess the levels of nucleotides derived from both de novo and salvage pathways.

However, the authors support their conclusions by reporting the increasing levels of nucleotide monophosphate, nucleosides, and nucleobases from 6 hours onwards. Additionally, it is noted a decrease in nucleotide diphosphates and triphosphates after 6 hours. This is indeed consistent with their hypothesis.

Then, they proceeded to investigate the potential impact of a pharmacological intervention on the savage pathway. Utilizing a replicon system, they explored the effects of inhibiting the purine salvage pathway. Specifically, they employed 6-mercaptopurine, a compound known to target HGPRT, an indispensable enzyme within the purine salvage pathway. At 4 hours post-infection (hpi), they observed a slight decline in RNA synthesis, equivalent to a reduction of roughly 5 to 8-fold in RNA accumulation. One issue with this result is that the inhibitor may have some pleiotropic effect resulting in reduction of virus replication. Is there any way to address this concern?

The authors then proceeded by integrating the metabolomics dataset with the phosphoproteomic data of CVB3-infected HeLa R19 cells. Through this intriguing analysis, they discovered five proteins implicated in nucleotide metabolism. Collectively, the integration of phosphoproteomic and metabolomic data suggests that nucleotide metabolism experiences alterations not only at the metabolome level but also at the phosphoproteome level. Inhibition of one of the enzymes discovered, AMPD2, had only a modest effect when inhibiting on CVB3 replication. This suggests that these enzymes are not essential for CVB3 replication.

This fails to provide a clear mechanistic understanding of how CVB3 and EMCV However, I am of the opinion that the study has been meticulously executed, with the discussion of the outcomes and conclusions lucidly articulated. The significance of this study lies in its ability to illuminate a crucial yet poorly understood facet of RNA virus replication.

Reviewer #2: The results are new and interesting, and the manuscript is well written. 

1. There is no statistical analysis on Figs 1,2, 3 and 4. There is no indication of the type of statistical analysis used for the data in Fig 3.

2. Phosphorylation of proteins must be confirmed by WB

3. The role of mTOR signaling must be confirmed by its pharmacological or genetic inhibition.

4. Does inhibition of nucleotide salvage reduces viral protein synthesis and progeny release?

Reviewer #3: 1. The metabolomics and tracing data are mostly performed in uninfected and infected HELA cells. HELA cells express high levels of Human papillomavirus E6 and E7 which are known to alter cellular metabolism and would mask many changes in metabolism that might be induced by picornaviruses.

2. They do not address why nucleobases, nucleosides, NMPs and NDPs are induced by infection but not NTPs in HELA cells. There is also limited discussion on the differences seen in HUH cells, namely that NTPs are upregulated only in HUH cells.

3. The salvage pathway does not appear to be strongly induced and if anything is flat or limited in the infected cells as compared to the uninfected cells as determined by the percentages of M+5 labeling. They state throughout that the salvage pathway is activated by infection which is not so clear in the data presented. They show inhibition of purine or pyrimidine salvage pathway has modest effects on replication which goes along with the limited activation. Additionally, they do not show production of virus, just increase in luciferase showing increased RNA replication of the virus. Importantly, they only show this modest effect at 4 hours post infection while the differences in purine and pyrimidine metabolism is mostly seen only at later times post infection.

4. While the combination of metabolomics and phosphoproteomics is potentially interesting, all of the phosphosite analysis is correlative and no effects of phosphorylation on purine or pyrimidine metabolism is shown in the infected cells. Also may of the phosphosites they discuss have unknown effects on the protein function so does not make a clear point. Furthermore, knockdown of the two of the enzymes that have altered phosphorylation on specific sites, has no effect on viral replication indicating that they are likely not relevant to the changes in infected purine and pyrimidine metabolism.

5. They draw many large conclusions that they do not go far enough experimentally to demonstrate in the manuscript

a. They state “We present evidence that both CVB3, an enterovirus, and EMCV, a cardiovirus, increase purine and pyrimidine nucleotide levels by promoting degradation of nucleic acids and nucleotides as well as triggering salvage pathways.” However, they do not show direct evidence that the degradation is activated and the data for the salvage pathway labeling is not as clear as they indicate as it appears to level off faster than in the uninfected cells indicating it is partially shut down not activated.

b. IN the discussion section, they extensively discuss RNA degradation pathways and their potential relevance to the increased purines and pyrimidines. However, they focus on enzymes like RNaseL and XRN/AUF, that are inhibited by picornaviruses which does not make sense for their hypothesis that RNA degradation is increased.

**Part III – Minor Issues: Editorial and Data Presentation Modifications**

Reviewer #1: (No Response)

Reviewer #2: (No Response)

Reviewer #3: (No Response)

PLOS authors have the option to publish the peer review history of their article (what does this mean?). If published, this will include your full peer review and any attached files.

Reviewer #1: No

Reviewer #2: **Yes: **Rodrigo Franco

Reviewer #3: No

Figure Files:

Data Requirements:

Reproducibility:

References:

---

## [Editor Report · Decision Letter 1]

8 Feb 2024

Dear Dr. Berkers,

We are pleased to inform you that your manuscript 'Modulation of nucleotide metabolism by picornaviruses' has been provisionally accepted for publication in PLOS Pathogens.

Best regards,

George A. Belov, PhD

Academic Editor

PLOS Pathogens

Guangxiang Luo

Section Editor

PLOS Pathogens

Michael Malim

Editor-in-Chief

PLOS Pathogens

orcid.org/0000-0002-7699-2064
---

## [Editor Report · Acceptance letter]

16 Feb 2024

Dear Professor Berkers,

We are delighted to inform you that your manuscript, "Modulation of nucleotide metabolism by picornaviruses," has been formally accepted for publication in PLOS Pathogens.

Best regards,

Michael Malim

Editor-in-Chief

PLOS Pathogens

orcid.org/0000-0002-7699-2064